# Rab5-mediated endosome formation is regulated at the *trans*-Golgi network

Makoto Nagano[1], Junko Y. Toshima[2]*, Daria Elisabeth Siekhaus [3] & Jiro Toshima[1]*

Early endosomes, also called sorting endosomes, are known to mature into late endosomes via the Rab5-mediated endolysosomal trafficking pathway. Thus, early endosome existence is thought to be maintained by the continual fusion of transport vesicles from the plasma membrane and the *trans*-Golgi network (TGN). Here we show instead that endocytosis is dispensable and post-Golgi vesicle transport is crucial for the formation of endosomes and the subsequent endolysosomal traffic regulated by yeast Rab5 Vps21p. Fittingly, all three proteins required for endosomal nucleotide exchange on Vps21p are first recruited to the TGN before transport to the endosome, namely the GEF Vps9p and the epsin-related adaptors Ent3/5p. The TGN recruitment of these components is distinctly controlled, with Vps9p appearing to require the Arf1p GTPase, and the Rab11s, Ypt31p/32p. These results provide a different view of endosome formation and identify the TGN as a critical location for regulating progress through the endolysosomal trafficking pathway.

[1] Department of Biological Science and Technology, Tokyo University of Science, 6-3-1 Niijyuku, Katsushika-ku, Tokyo 125-8585, Japan. [2] School of Health Science, Tokyo University of Technology, 5-23-22 Nishikamada, Ota-ku, Tokyo 144-8535, Japan. [3] Institute of Science and Technology Austria, Am Campus 1, A-3400 Klosterneuburg, Austria. *email: toshimajk@stf.teu.ac.jp; jtosiscb@rs.noda.tus.ac.jp

Endocytosis allows the cell to internalize various cargos, such as membrane proteins and extracellular molecules, into the cell through vesicles that bud off from the plasma membrane (PM). Once endocytic vesicles are internalized into the cytosol, they are rapidly targeted to the early endosome (EE) that is the primary sorting station from which endocytosed materials can be recycled back to the PM[1], or brought to late endosomes (LEs) en route to the lysosome/vacuole for degradation[2]. In mammalian cells, EEs, also known as sorting endosomes, are composed of thin tubular regions and large vesicular regions containing membrane-bound intraluminal vesicles[3]. These morphologically distinct structures in the EE are functionally important, and the tubular regions contain proteins targeted for the recycling pathway whereas large vesicular regions are involved in sorting to the LE[4]. These EEs exist as stable structures, but their existence is thought to be maintained by continual vesicular transport from the PM or other organelles, because a whole or part of them converts into LEs/multivesicular bodies[5]. However, whether either or both of these transport pathways is necessary for the formation of EEs and LEs has not been clarified.

While EEs are well characterized in mammalian cells[6], whether those structures exist in yeast is still controversial. The budding yeast *Saccharomyces cerevisiae* is an important organism for studies of endocytic mechanisms due to its many experimentally advantageous properties[7]. Several studies demonstrated that *S. cerevisiae* has at least two types of endosomes, early-stage endosomes that contain the Rab5 homolog Vps21p[8] and late-stage endosomes that contain the Rab7 homolog Ypt7p[9]. Since Rab5 and Rab7 GTPases are key determinants of EEs and LEs in many organisms[10], *S. cerevisiae* seems to contain EE-like organelles, but a recent study demonstrated that the yeast *trans*-Golgi network (TGN) serves the role of an early and recycling endosome in the endocytic pathway of *S. cerevisiae* and that distinct EEs do not exist[11]. The TGN is a major sorting station in the secretory pathway that directs newly synthesized proteins to different subcellular destinations, such as the PM, endosome, and lysosome/vacuole[12,13]. The TGN also receives endocytosed proteins from the EE or LE through a retrograde route, and recycles back them to the PM[12,14]. In addition to these conventional roles, the TGN directly fuses with endocytic vesicles[11]. In contrast, other studies, using fluorescent markers of the endocytic pathway, demonstrated the existence of distinct EEs that are highly motile and associate with endocytic vesicles[15,16]. It was also recently reported that yeast has a recycling route that directly transports endocytosed cell surface membrane proteins from EEs to the cell surface[17]. These contradictory observations make it difficult to understand how endosomes are formed and maintained in yeast.

The Rab5 GTPase has been proposed to be a master regulator of endosome biogenesis and trafficking[18–20], playing a key role in the maturation of the early to the late endosome[21–23]. This maturation process is regulated by a sequential shift of activity from the early endosomal Rab5 to the late endosomal Rab7, a process termed Rab conversion[21,22]. In general, Rab conversion is mediated by Guanine nucleotide exchange factors (GEFs), and an upstream Rab recruits a GEF for a downstream Rab[24,25]. During early to late endosome maturation, Rab5 recruits the Mon1–Ccz1 complex, a GEF for Rab7, and promotes Rab5–Rab7 conversion; this mechanism is conserved in several organisms including *S. cerevisiae*[22,26,27]. Although Rab5-to-Rab7 conversion is well characterized, which Rab protein in the endocytic pathway lies upstream of Rab5 activation has not been clarified.

To understand the full mechanism regulating Rab5 activity we first need to identify on which organelle the GEF for Rab5 is first localized and how this recruits Rab5 to the EE. The Vps9 domain-containing proteins, Vps9p (yeast Rabex-5) and Muk1p have been identified as specific GEFs for yeast Rab5s[28,29]. Vps9p can be recruited to endosomes through the interaction of its CUE domain with ubiquitinated endocytic cargo proteins, thereby promoting the endosomal localization of the yeast Rab5, Vps21p[30]. However, the finding that non-ubiquitinated cargo also requires Vps21p for its transport from the TGN to the vacuole in the VPS pathway[31], suggests that an ubiquitin and endocytic pathway-independent mechanism exists to recruit Vps21p to the EE. We recently demonstrated that the endocytic pathway intersects with the pathway from the TGN at an early stage of endocytosis, independently of yeast Rab5s[15]. After the convergence of these two pathways, Vps21p is recruited to the endosome and functions to promote the subsequent endocytic processes[15], therefore, traffic from the TGN might have some role in triggering Vps21p activation by an as yet unknown mechanism.

In the present study, we show that endocytic vesicle internalization is not essential, but that vesicle transport from the TGN is crucial for Vps21p-mediated endosome formation. We also demonstrate that the Vps21p-GEF Vps9p is first recruited to the TGN dependent on the Arf1 GTPase, and then transported to the endosome to activate Vps21p through Ent3p/5p-mediated vesicle transport. We further show that the yeast Rab11s, Ypt31p/32p, regulate recruitment of the epsin-related adaptors Ent3p and Ent5p to the TGN, and thus these Rab proteins are required for Vps21p activation and therefore upstream of Rab5 activation. These findings demonstrate the importance of post-Golgi transport in Vps21p-mediated endosome formation and trafficking.

## Results

**Endocytosis is not essential for the endosome formation.** Rab5 is a key regulator of endosome fusion and trafficking, but whether endocytotic vesicle internalization is necessary for Rab5 function has not been determined. To clarify this, we utilized two yeast mutants, *sla2Δ* and *sac6Δ scp1Δ* mutants, which have defects in clathrin-mediated endocytosis[32,33], and a control strain lacking the Vps21p GEFs, Vps9p, and Muk1p, which regulate the endosomal localization of Vps21p[28]. We first confirmed the endocytic defect in these mutants using Alexa Fluor 594-labeled yeast mating pheromone α-factor (A594-α-factor), a marker of endocytosis[34,35]. Consistent with previous reports[32,33], *sla2Δ* and *sac6Δ scp1Δ* mutants exhibited severe defects in the uptake of A594-α-factor from the PM, while in wild-type cells the majority of A594-α-factor was transported to the vacuole by 20 min after addition (Fig. 1a, b). In the *vps9Δ muk1Δ* mutant, Alexa-α-factor was internalized normally but accumulated in multiple endosomal compartments at 20 min after addition (Fig. 1a, b), suggesting a delay of α-factor transport to the vacuole after internalization. To further confirm the endocytic defect, we next examined the internalization of 3-triethylammoniumpropyl-4-p-diethylaminophenylhexatrienyl pyridinium dibromide (FM4-64), a lipophilic styryl dye that is used to follow bulk membrane. When added to wild-type cells, FM4-64 is immediately incorporated into the PM and internalized via bulk-phase endocytosis, and then transported to the vacuole within 20 min (Supplementary Fig. 1a, b). Similar to Alexa-α-factor uptake, the *sac6Δ scp1Δ* mutant showed a remarkable defect in the uptake of FM4-64 from the PM and the *vps9Δ muk1Δ* mutant showed a delay of FM4-64 movement to the vacuole after internalization (Supplementary Fig. 1a, b). Since yeast also has a clathrin-independent endocytic pathway that depends on the Rho1 GTPase[36], we generated a triple mutant lacking the Rho1-GEF, Rom1p, in addition to Sac6p and Scp1p. The *sac6Δ scp1Δ rom1Δ* triple mutant exhibited a more severe defect in FM4-64 uptake from the PM, compared to the *scp1Δ rom1Δ* mutant (Supplementary Fig. 1a, b). We next examined the localization of Vps21p in these mutants. To precisely evaluate

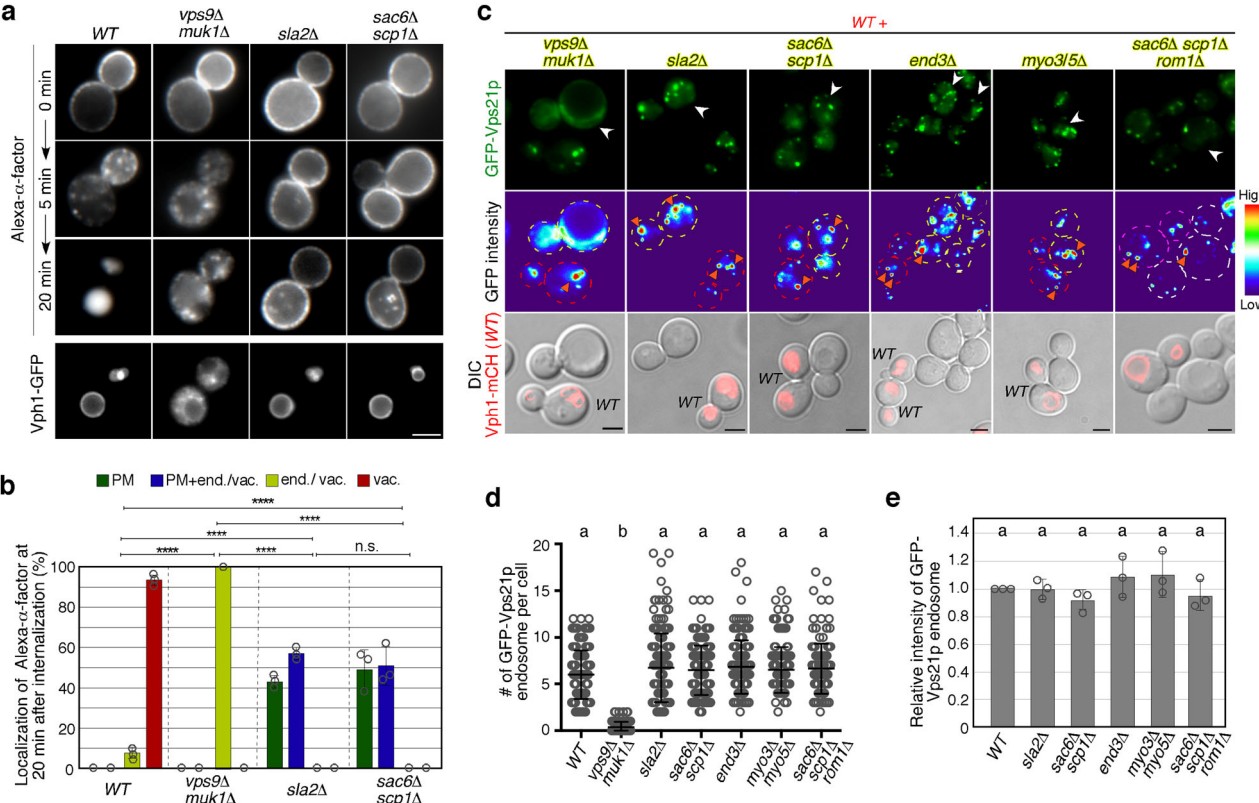

**Fig. 1** Defective endocytosis does not affect Vps21p-mediated endosome formation. **a** Effect of the deletion of Rab5-specific GEFs or endocytosis-related proteins on the internalization of Alexa Fluor594-labeled α-factor (Alexa-α-factor) or Vhp1-GFP transport to the vacuole. The images were acquired at 0, 5, and 20 min after washing out unbound Alexa-α-factor (Alexa-α-factor). **b** Quantification of the intracellular compartments accumulating Alexa-α-factor in the indicated cells at 20 min after internalization. The compartments were categorized into four classes; plasma membrane only (PM), PM and endosome and/or vacuole (PM + end./vac.), endosome and/or vacuole (end./vac.), and vacuole only (vac.). **c** Localization of GFP-Vps21p in wild-type (*WT*) and mutant cells. *WT* and mutant cells expressing GFP-Vps21p were grown to early-logarithmic to mid-logarithmic phase, mixed, and acquired in the same images. Fluorescence images or heat maps showing GFP levels are shown in the panels labeled GFP-Vps21p or GFP intensity, respectively. *WT* or mutant cells are indicated with red or yellow dashed lines, respectively. *WT* cells are labeled by the expression of Vph1-mCherry (red) which is shown in the lower images overlaid with DIC images. **d**, **e** Quantification of the (**d**) number or (**e**) fluorescence intensity of GFP-Vps21p-positive endosomes displayed in (**c**). Data show mean ± SEM from at three independent experiments, (**b**) with 50 cells or (**e**) 100 endosomes, or (**d**) mean ± SD with 150 cells. *$p < 0.05$, ***$p < 0.001$, ****$p < 0.0001$, n.s., not significant, chi-square test for trend (**b**). Different letters indicate significant difference at $p < 0.0001$, one-way ANOVA with Tukey's post-hoc test (**d**, **e**). Scale bar in all panels, 2.5 μm

differences in the localization of Vps21p, each mutant was compared directly alongside wild-type cells that were labeled by their expression of Vph1p-mCherry (Fig. 1c). We previously demonstrated that Vps21p is predominantly localized at the EE-to-LEs, and little localized at the TGN[15]. Consistent with this observation, in the wild-type cell GFP-Vps21p was localized at multiple endosomal compartments, whereas deletion of the *VPS9* and *MUK1* genes led to the complete relocalization of Vps21p to the cytosol (Fig. 1c, d). In contrast, the *sla2Δ* and *sac6Δ scp1Δ* mutants exhibit Vps21p localization similar to that in wild-type cells (Fig. 1c). The *sac6Δ scp1Δ rom1Δ* triple mutants also exhibited Vps21p localization similar to that in wild-type cells (Fig. 1c). By comparing the localization of GFP-Vps21p with tdTomato-tagged Hse1p, a marker of the EE-to-LEs[37,38], we confirmed that GFP-Vps21p localizes at endosomal compartments in *sac6Δ scp1Δ* cells, similar to wild-type cells (Supplementary Fig. 1c, d), as described previously[15]. Quantitative analysis revealed that the number of GFP-Vps21p-labeled endosomes in the endocytosis-defective mutants is almost the same as that in wild-type cells (Fig. 1d). The fluorescence intensity of GFP-Vps21p on endosomes was also not significantly changed in the endocytosis-defective mutants, compared to wild-type cells (Fig. 1e). Similar results were obtained when we used other

endocytic mutants, such as *end3Δ* and *myo3Δ myo5Δ* (Fig. 1d, e). Since Vps21p functions in the VPS pathway from the TGN to the vacuole, as well as the endocytic pathway, we examined if the vacuolar pathway is intact in the endocytic mutants by using Vph1-GFP as a marker[15,31]. As expected, in the *vps9Δ muk1Δ* mutant that blocks the VPS pathway, Vph1-GFP accumulated in multiple puncta, similarly to what is seen in the *vps21Δ* mutant[15]; however in the *sla2Δ* and *sac6Δ scp1Δ* mutants it is normally transported to the vacuole (Fig. 1a). The number and fluorescence intensity of GFP-Vps21p-labeled endosomes was also not significantly changed in cells treated with Latrunculin A, which abolishes both clathrin-dependent and clathrin-independent endocytosis (Supplementary Fig. 1e–g)[36,39]. Additionally, we found that the number and fluorescence intensity of structures labeled with Vps8-GFP, which is a marker for the late or pre-vacuolar endosome[11], are not changed in the *sac6Δ scp1Δ* mutants (Supplementary Fig. 1h–j). These results indicated that Vps21p is normally localized and functions at the endosome in these endocytic mutants, and thus, endocytic internalization is not essential for Vps21p-mediated endosome formation.

**Post-Golgi transport is required for the endosome formation.** We wished to determine if an alternative trafficking pathway

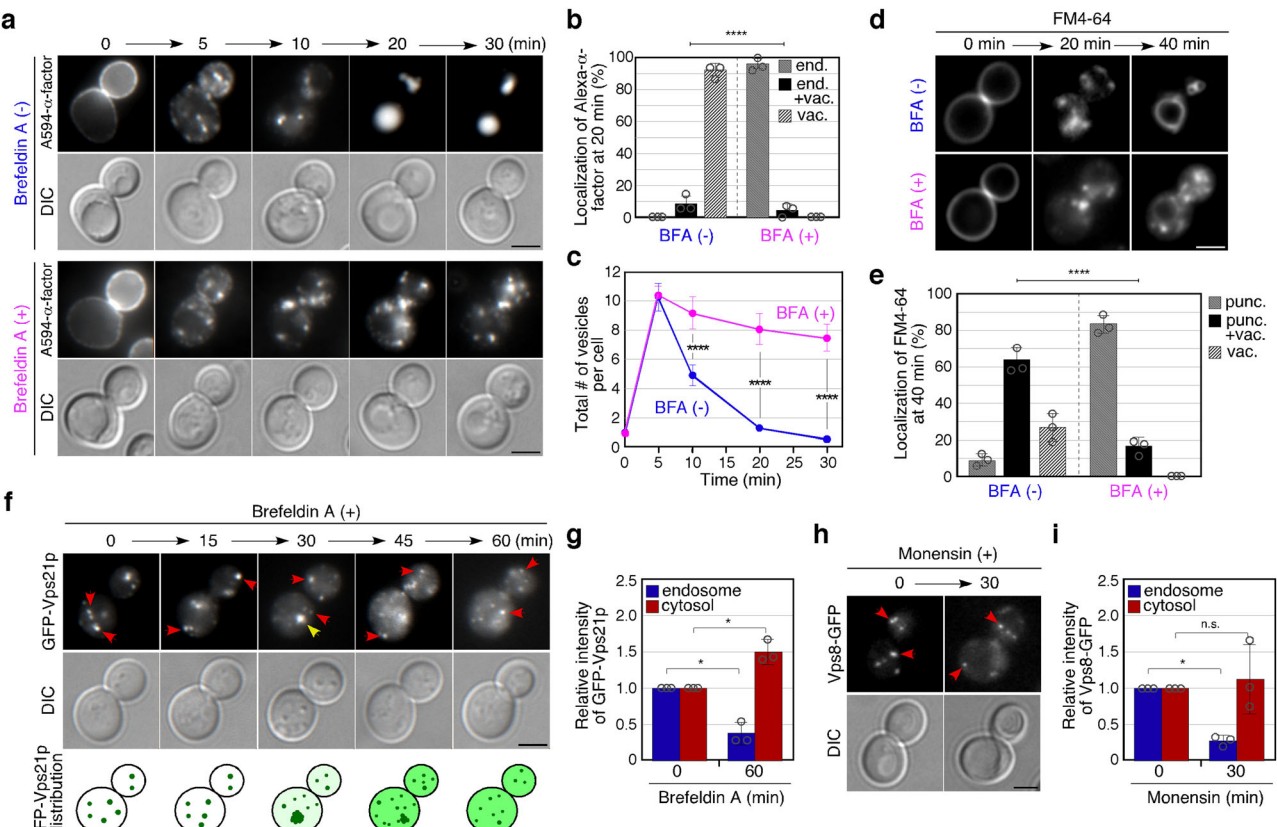

**Fig. 2** The effect of inhibiting post-Golgi traffic on Vps21p-mediated vesicle formation and trafficking. **a** The spatio-temporal localization of Alexa-α-factor in Brefeldin A-treated cells. Cells were labeled with Alexa-α-factor in the presence or absence of 100 µg m$^{-1}$ L$^{-1}$ Brefeldin A (BFA). The images were acquired at the indicated time after internalization of Alexa-α-factor. **b** Quantification of Alexa-α-factor localization in the cells at 20 min after internalization. Endosome only (end.), endosome and vacuole (end. + vac.) and vacuole only (vac.). **c** Quantification of the number of Alexa-α-factor-positive vesicles displayed in **a**. **d** Effect of BFA treatment on FM4-64 transport from the PM to the vacuole. After treatment of the cells with 100 µg m$^{-1}$ L$^{-1}$ BFA for 15 min, cells were labeled with 200 µM FM4-64 for 15 min on ice and observed at 0, 20, and 40 min after washing out unbound FM4-64 and incubating the cells at 25 °C. **e** Quantification of FM4-64 localization in the cells at 40 min after internalization. Puncta only (punc.), puncta and vacuole (punc. + vac.), and vacuole only (vac.). **f** The effect of BFA on the localization of Vps21p. Cells expressing GFP-Vps21p were incubated with 100 µg m$^{-1}$ L$^{-1}$ BFA at 25 °C and observed at the indicated time after the incubation. Red arrows indicate the example of GFP-Vps21p endosomes and yellow arrow shows aberrant accumulation of GFP-Vps21p. **g** The graph shows quantification of the fluorescence intensity of GFP-Vps21p at the endosomes and in the cytoplasm. **h** The effect of Monensin on the localization of Vps8p. Cells expressing Vps8-GFP were incubated with 50 µM Monensin at 25 °C and observed at the indicated time after the incubation. **i** The graph shows quantification of the fluorescence intensity of Vps8-GFP at the endosomes and in the cytoplasm. Data show mean ± SEM from three independent experiments, with 50 cells **b**, **c**, 100 cells **e** or 100 endosomes **g**, **i**. ****$p < 0.0001$, chi-square test for trend **b**, **e**, two-way ANOVA with Bonferroni's post-hoc test **c**. *$p < 0.05$, unpaired $t$-test with Welch's correction **g**, **i**. Scale bar in all panels, 2.5 µm

could fuel endosome formation and focused on the VPS pathway from the TGN which converges with the endocytic pathway at an early stage of endocytosis, independently of yeast Rab5s[15]. We first examined the requirement for vesicle transport from the TGN for Vps21p activation. We utilized two major drugs, Brefeldin A (BFA) and Monensin, which perturb post-Golgi transport by inhibiting ER–Golgi or intra-Golgi traffic[40,41]. Intriguingly, treatment of wild-type cells with BFA or Monensin caused similar defects in Alexa-α-factor transport to those seen in the $vps9\Delta$ $muk1\Delta$ or $vps21\Delta$ mutant:[15] Alexa-α-factor was internalized normally but accumulated in multiple dots, localizing GFP-Vps21p, at 20–30 min after addition (Fig. 2a, Supplementary Figs. 2a, and 3a). In BFA-untreated or Monensin-untreated cells, colocalization of Alexa-α-factor with GFP-Vps21p-labeled dots is only transient at 5–10 min after internalization but in BFA-treated or Monensin-treated cells the colocalization was still maintained after 10–30 min (Supplementary Figs. 2c and 3d), suggesting that these drug treatments might inhibit the process of endosome fusion mediated by Vps21p. By comparing the localization of GFP-Vps21p with Sec7p or Hse1p, markers for the TGN

or endosomes[15], respectively, we found that Vps21p is localized at endosomal compartments in BFA-treated cells, as well as in untreated cells (Supplementary Fig. 2d, e). Quantitative analysis categorizing the Alexa-α-factor localization as endosome only, endosome and vacuole, or vacuole only, revealed that these drugs inhibit the transport of Alexa-α-factor at the Vps21p-residing endosome (Fig. 2b, Supplementary Fig. 3b). We further analyzed temporal changes in the number of Alexa-α-factor-labeled endosomes. In wild-type cells, the number increased by 5 min, and then rapidly decreased until most Alexa-α-factor had been transported to the vacuole (Fig. 2c, Supplementary Fig. 3c). In contrast, the number of Alexa-α-factor-labeled endosome did not substantially change after reaching a maximum in BFA or Monensin-treated cells (Fig. 2c, Supplementary Fig. 3c). We observed a similar delay upon utilizing FM4-64 in BFA-treated or Monensin-treated cells (Fig. 2d, e, Supplementary Fig. 3e, f). Since changes in the number of A594-α-factor-labeled endosomes in cells treated with these drugs were quite similar to those in $vps21\Delta$ $ypt52\Delta$ cells[15], we speculated that these drugs might affect the activation status of Vps21p. To examine this, we observed the

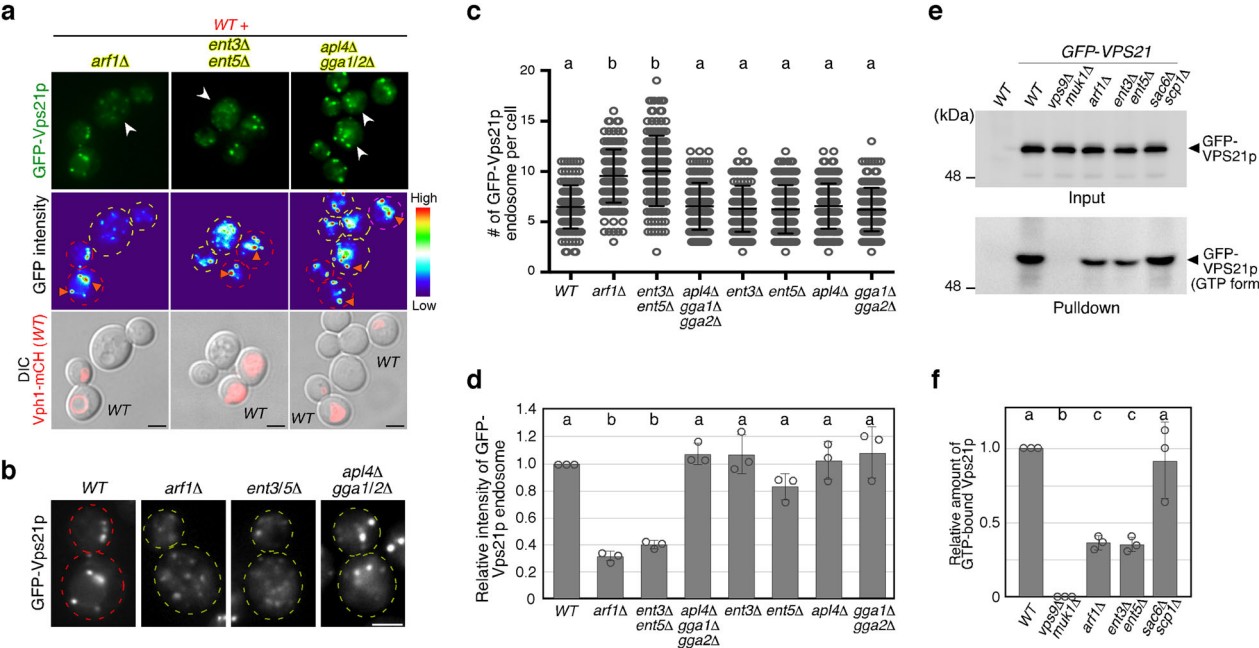

**Fig. 3** The effect of deleting Arf1p or adaptor proteins on the localization and activity of Vps21p. **a**, **b** The localization of GFP-Vps21p in wild-type (*WT*) and mutant cells. Fluorescence images were acquired as shown in Fig. 1c. High magnification images indicated by arrowhead in **a** were shown in **b**.
**c** Quantification of the number of GFP-Vps21p-positive endosomes displayed in **a** and Supplementary Fig. 2a. **d** Fluorescence intensity of GFP-Vps21p-positive endosomes displayed in **c**. **e** Immunoblots showing active levels of Vps21p. Endogenous Vps21p was tagged with GFP in the indicated cells, and 3 μg of total cell lysate (2% input) were loaded and immunoblotted with an anti-GFP antibody (Input panel). Active Vps21p from 150 μg of total cell lysate was pulled down with GST-tagged N terminal portion of human EEA1 (GST-EEA1NT) and probed with an anti-GFP antibody (Pulldown panel).
**f** Quantification of active Vps21p levels displayed in **e**. Graph shows mean ± SEM from three independent experiments. Data show mean ± SD with 150 cells **c** or mean ± SEM with 100 endosomes **d** from three independent experiments. Different letters indicate significant difference at *p* < 0.05, one-way ANOVA with Tukey's post-hoc test **c**, **d**, **f**. Scale bar, 2.5 μm. Uncropped blots for **e** can be found in Supplementary Fig. 11

localization of Vps21p as an indirect readout of its activity because the GTP-bound active form is targeted to the endosomal membranes and the GDP-bound inactive form is localized in the cytosol[42,43] (Supplementary Fig. 4a). As expected, treatment of wild-type cells with these drugs changed the Vps21p localization from the endosome to the cytosol in a time-dependent manner (Fig. 2f and Supplementary Fig. 4b). We observed that Vps21p transiently accumulates (Fig. 2f and Supplementary Fig. 4c, 5–30 min), and then gradually disperses in the cytosol (Fig. 2f and Supplementary Fig. 4c, 10–60 min). We note that BFA treatment changed Arf1p localization within 5 min, and also induce accumulation of Vps21p with the same timing (Supplementary Fig. 4c). At 60 min (BFA) or 30 min (Monensin) after drug treatment, the intensity of GFP-Vps21p at endosomes decreased to ~37% or ~23% of that in the wild-type cells and increased in the cytosol to ~1.5 or ~1.9 fold that of the wild-type cells in the BFA-treated or Monensin-treated cells, respectively (Fig. 2g and Supplementary Fig. 4d). The intensity of Vps8-GFP at endosomes also decreased to ~27% at 30 min after Monensin treatment (Fig. 2h, i). These observations suggest that vesicle transport from the TGN is important for Vps21p-mediated endosomal transport to the vacuole.

**Arf1p and Ent3p/5p are required for Vps21p activation.** We sought to identify Golgi-resident proteins that are required for the regulation of Vps21p activity on the endosome. Arf1p has been demonstrated to be a key regulator of Golgi-to-vacuole transport[44]. Thus, we next examined the effect of deleting the *ARF1* gene on Vps21p. We first examined if GFP-Vps21p localizes at the endosome in the *arf1Δ* mutant by comparing the localization with Hse1-tdTomato. We found that GFP-Vps21p dots in the

*arf1Δ* mutants colocalize with Hse1-tdTomato, similar to what is seen in wild-type cells (Supplementary Fig. 5a, b). In the *arf1Δ* mutant, the number of the endosomes containing GFP-Vps21p was increased (Fig. 3a–c), whereas the fluorescence intensity at the endosomes was decreased relative to that in wild-type cells (Fig. 3a, d), suggesting that Vps21p-mediated endosomal transport is impaired in this mutant. To further confirm the requirement for post-Golgi transport in endosomal transport, we utilized cells lacking the clathrin adaptor proteins that regulate transport from the TGN to the endosome. In *S. cerevisiae*, three classes of TGN-resident adaptors, the AP-1 complex, the Gga homolog Gga1/2p and epsin-related Ent3p/5p, have been identified[44,45]. By expressing GFP-Vps21p in cells lacking single or multiple adaptors, we found that a double deletion of *ENT3* and *ENT5* genes causes relocalization of much of the endosomal GFP-Vps21p to the cytosol, similar to BFA-treated cells or the *arf1Δ* mutant (Fig. 3a–d and Supplementary Fig. 5c). GFP-Vps21p dots in the *ent3Δ ent5Δ* mutant also colocalized with Hse1-tdTomato (Supplementary Fig. 5a, b), indicating that these dots are endosomes. In the *ent3Δ ent5Δ* double mutant, the fluorescent intensity of GFP-Vps21p-labeled endosomes was significantly decreased (Fig. 3d), whereas the number of the endosomes was increased (Fig. 3c). These observations suggest a possibility that decreased endosomal localization of activated Vps21p reduces the competence of endosomal transport, causing increase of punctate compartments labeled with GFP-Vps21p in this mutant. In contrast, deletion of Apl4p, the γ-subunit of the AP-1 complex, Gga1/2p, or even a triple deletion of these adaptors, had a negligible effect on the number and fluorescent intensity of Vps21p endosome (Supplementary Fig. 5c). These results, therefore, support the idea that Arf1p-mediated and Ent3p/

Ent5p-mediated post-Golgi transport is important for Vps21p-mediated endosomal transport.

We further investigated if post-Golgi transport is required for the activity of Vps21p using a pull-down assay. The active forms of Rab5A/B have been shown to interact directly with the N-terminal region (1-209 a.a.) of human EEA1 (EEA1NT)[46] (Supplementary Fig. 6a). We purified GST-fused EEA1NT (Supplementary Figs. 6a, b, 10) and confirmed that it specifically binds to the GTP-bound form of Vps21p (Vps21$^{Q66L}$), but not the GDP-bound form of Vps21p (Vps21$^{S21N}$) (Supplementary Fig. 6c). Using GST-EEA1NT, we next tested the amount of the active GTP-bound form of Vps21p in wild-type cells and in the mutants that affect post-Golgi transport. We first examined the functionality of the assay, and showed that Vps21p was efficiently pulled down from the cell lysate prepared from wild-type cells, but rarely from extract lacking the Vps21p-GEFs (Fig. 3e, f), indicating that GST-EEA1NT specifically binds to the active Vps21p. Consistent with the observed normal Vps21p localization at endosomes in the sac6Δ scp1Δ mutant (Fig. 1c–e), a similar amount of Vps21p was pulled down from this double mutant (Fig. 3e, f) as from the wild-type cells. In contrast, the active Vps21p was significantly decreased in arf1Δ or ent3Δ ent5Δ mutants to $37 \pm 5\%$ or $36 \pm 5\%$ of the wild-type cells, respectively (Fig. 3e, f). Furthermore, expression of the GTP-bound form of Vps21p partially suppresses the growth defect of the arf1Δ mutant at 37 °C (Supplementary Fig. 6d). Thus, Arf1p-mediated and Ent3p/5p-mediated post-Golgi transport seems to be important for the Vps21p activation.

**Arf1p and Ent3/5p are required for Vps9p localization.** We wondered if the importance of post-Golgi transport for Vps21p activity might be due to an effect on Vps9p. Vps9p was found at similar levels in wild-type and in arf1Δ or ent3Δ ent5Δ mutant cells (Fig. 4a) so we focused on its localization. Live-cell imaging of GFP-Vps9p revealed Vps9p at several puncta, in addition to the cytosol in wild-type cells (Fig. 4b), as reported previously[28]. Interestingly, we found that the number of puncta containing GFP-Vps9p (Fig. 4c) and its residence time at the puncta (Fig. 4d–f, and Supplementary Movie 1) were increased relative to wild-type in the ent3Δ ent5Δ mutant. To make Vps9p localization clearer, we expressed GFP-Vps9p under the control of the ZWF1 gene promoter, which moderately increased its expression, compared with the authentic promoter (Supplementary Figs. 7a and 10)[47]. We obtained similar results showing increased Vps9p puncta and increased residence time of Vps9p at the puncta in the ent3Δ ent5Δ mutant (Supplementary Fig. 7b–f). Through comparisons with Sec7p or Hse1p, we found that Vps9p is predominantly localized at the endosomes in wild-type cells, but that deletion of the ENT3 and ENT5 genes significantly increased Vps9p's localization at the TGN and decreased it at the endosomes (Fig. 4g, h). Taken together with the observations that Vps21p is localized to the cytosol and displays a decreased activity in the ent3Δ ent5Δ mutant, these results suggest that decreased localization of Vps9p at the endosomes might affect the activity of Vps21p in the ent3Δ ent5Δ mutant.

We also wished to examine the effect of ARF1 gene deletion on Vps9p localization. Deletion of the ARF1 gene also impaired the proper localization and activation of Vps21p (Fig. 3), but we could not precisely assess the effect on Vps9p's TGN localization because of the high fluorescence intensity in the cytosol (Fig. 4b). Previous studies demonstrated that Vps9p accumulates at aberrant endosomes deemed class E compartment in cells lacking Vps4p, which catalyzes the release of the ESCRT complex from the endosomal membrane[28,30]. Since in vps4Δ cell the fluorescence intensity of Vps9p in the cytosol was low enough to assess

the intensity at the endosomal compartments, we examined the effect of the ARF1 gene deletion on Vps9p localization, using this mutant. We observed the accumulation of GFP-Vps9p at the prevacuolar endosomal compartment in the vps4Δ mutant, which was decreased upon the additional deletion of the ARF1 gene (Supplementary Fig. 7g, h). This suggests a role for Arf1p in Vps9p recruitment to endosomal compartments. In addition, we utilized the BioID assay to examine the interaction between Arf1p and Vps9p. We fused bacterial biotin ligase BirA (R118G) mutant (BirA*) to Arf1p and expressed this hybrid protein to be able to biotinylate endogenous proteins that interact with Arf1p. Pull-down analysis with Streptavidin-agarose demonstrated that BirA*-tagged Arf1p could biotinylate Vps9p in vivo (Fig. 4i). These results are consistent with a potential role for Arf1p in the recruitment of Vps9p to the TGN before its transport to the endosome where it catalyzes nucleotide exchange on Vps21p.

**Ypt31p/32p are required for Vps9p transport to the endosome.** Recent studies have reported a functional relationship between Arf1p and Ypt31p/32p (yeast Rab11) at the trans-Golgi/TGN[48], thus we next examined if Ypt31p/32p play a role in Vps9p-mediated Vps21p activation. In wild-type cells, Vps9p was minorly localized at the TGN labeled by Sec7-mCherry and highly localized at endosomes labeled by Hse1-tdTomato (Fig. 5a, b). Interestingly, the TGN localization of Vps9p increased, and the endosome localization decreased in the ypt31(K127N) ypt32Δ temperature-sensitive (ypt31ts) mutant[49] at the non-permissive temperature (37 °C) (Fig. 5a, b and Supplementary Fig. 8a). The localization of Vps9p in the ypt31ts mutant at 37 °C was similar to that in the ent3Δ ent5Δ mutant (Fig. 4g, h), and this motivated us to further examine the localization of Vps21p and Ent3/5p in the ypt31ts mutant. As expected, at 37 °C the number of GFP-Vps21p-positive endosomes was significantly increased (Fig. 5c–e), whereas the fluorescence intensity at the endosomes was decreased in the ypt31ts mutant, compared to that in wild-type cells (Fig. 5f). Furthermore, we observed that in the ypt31ts mutant Ent3p and Ent5p clearly change their localization to the cytosol at 37 °C, although they are normally localized at the TGN at 25 °C (Fig. 5g). Quantitative analysis revealed that the fluorescent intensities of Ent3p or Ent5p at the TGN labeled by Sec7-GFP significantly decreases at 37 °C compared to that seen there at 25 °C (Fig. 5g, h). The fluorescent intensity of mCherry-tagged Apl2p, a β-subunit of AP-1 complex, decreases at 37 °C compared to that seen there at 25 °C, but this change seems not to be significant because the intensity of Sec7-GFP also decreases in the ypt31ts mutant at 37 °C (Fig. 5h). These results suggest a requirement for Ypt31p/32p function for the recruitment of Ent3/5p to the TGN, and subsequent Vsp9p-mediated Vps21p activation.

We wished to further confirm that Ypt31p/32p are needed for Vps21p activity. We first tested if Vps21p's cellular localization in another condition similarly requires Ypt31p/32p. Upon the deletion of the yeast Rab5-specific GAP Msb3p, Vps21p accumulates on the vacuolar membrane, due to being fixed into its GTP-bound form[20,43]. We observed this also in the ypt31ts mutant at 25 °C, but at 37 °C when Ypt31p function is inhibited, Vps21p accumulation was significantly diminished (Supplementary Fig. 8b, c). We then wished to test the effects of altering Ypt31p more subtly, altering its nucleotide state rather than its entire functionality. We tested Trs120p, a specific subunit of the TRAPPII complex that is reported to function as a GEF for Ypt31p[50]. Indeed, we found that membrane-bound Ypt31p is decreased in trs120 knockdown (kd) cells (Supplementary Fig. 8d) and that expression of a GTP-bound form of Ypt31p rescues the defective growth of trs120kd cells (Supplementary Fig. 8e). In

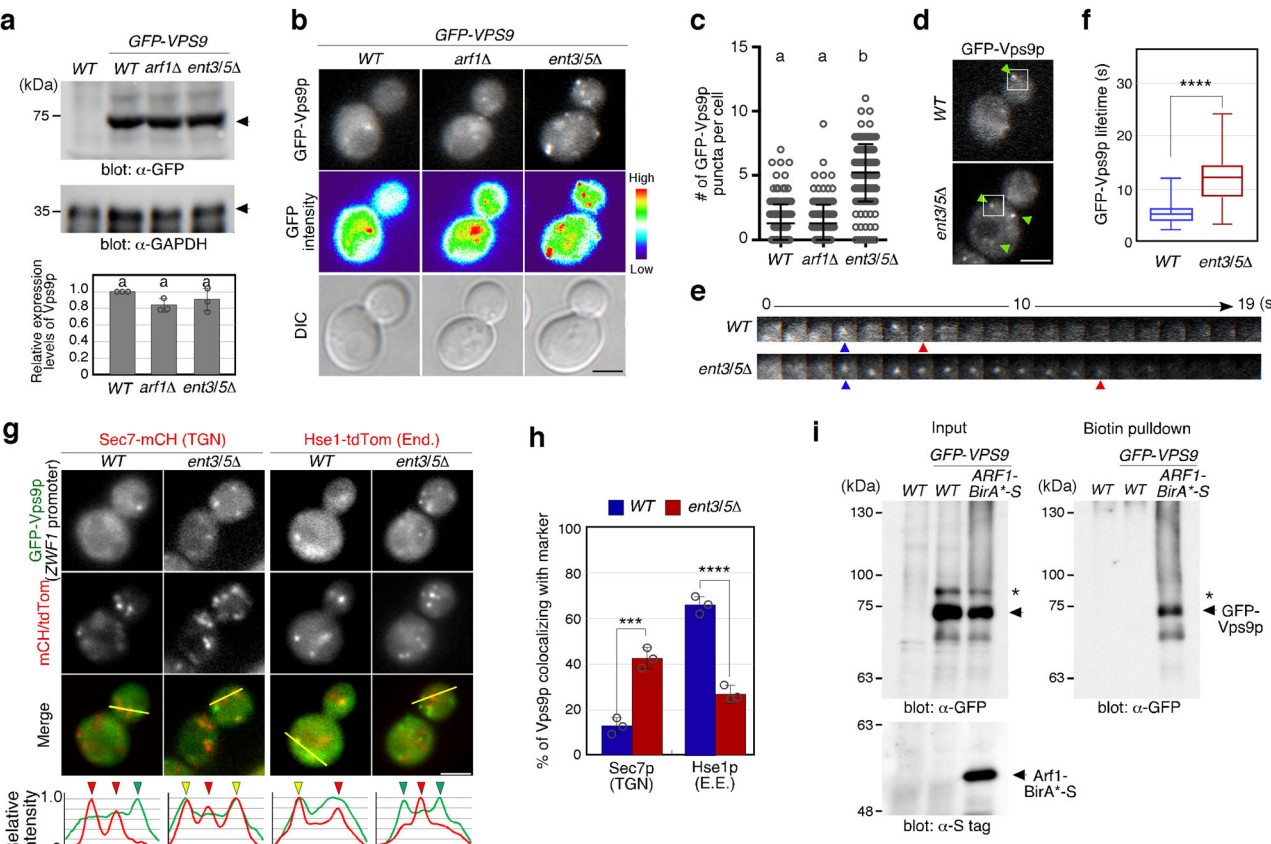

**Fig. 4** Arf1p and Ent3p/5p-dependent localization of Vps9p. **a** Immunoblots showing the expression levels of GFP-tagged Vps9p in the cells. GFP-Vps9p was expressed under the control of the authentic promoter from the endogenous locus. Total cell lysates were loaded and immunoblotted with an anti-GFP antibody (α-GFP panel). GAPDH was used as a loading control (α-GAPDH panel). Graph shows mean ± SEM from three independent experiments. **b** Localization of GFP-Vps9p in the cells. Fluorescence images (GFP-Vps9p), heat maps showing GFP levels (GFP intensity) and DIC images (DIC) are shown. **c** Quantification of the number of GFP-Vps9p puncta displayed in **b**. **d**, **e** Dynamic behavior of GFP-Vps9p puncta in the cells. Time series of the regions in the boxed area indicated in **d**. Blue and red arrowheads indicate appearing and disappearing points of GFP-Vps9p. **f** Graph shows the GFP-Vps9p lifetime in the cells. $n = 100$ puncta. Top and bottom bars are the 95% confidence limits. Data show the mean ± SEM of three independent experiments, with 100 cells. **g** Colocalization of GFP-Vps9p and Sec7p-mCherry (TGN) or Hse1p-tdTomato (Endosome; End) in the cells. Representative intensity profiles of GFP-Vps9p and Sec7-mCherry or Hse1-tdTomato along the yellow line in the merged images are indicated in the lower graphs. Yellow or red/green arrowheads indicate the presence or absence of colocalization, respectively. **h** The percentages of colocalization were calculated as the ratio of mCH/tdTom-tagged marker ($n = 100$) colocalizing with GFP-Vps9p-positive puncta. **i** Analysis of the interaction between Arf1p and Vps9p using the BioID assay. Vps9p and Arf1p were tagged with GFP and BirA*-S, respectively. Total cell lysate (1% input) or biotinylated proteins were loaded and immunoblotted with an anti-GFP (α-GFP panel) or anti-S-tag antibody (S-tag panel). Data show mean ± SD with 150 cells **c** or mean ± SEM with 100 puncta **f**, **h** from three independent experiments. Different letters indicate significant difference at $p < 0.05$, one-way ANOVA with Tukey's post-hoc test **a** and **c**. ****$p < 0.0001$, unpaired $t$-test with Welch's correction **f**. ***$p < 0.001$, ****$p < 0.0001$, two-way ANOVA with Tukey's post-hoc test **h**. Scale bar in all panels, 2.5 μm. Uncropped blots for **a** and **i** can be found in Supplementary Fig. 11.

*trs120kd* cells, Vps21p was partly dispersed in the cytosol, similar to what we observed in the *ypt31ts* mutant (Supplementary Fig. 8f). In *msb3Δ* cells, the *trs120kd* also shifted some Vps21p from the vacuolar membrane to the cytosol (Supplementary Fig. 8g). These observations support our conclusion that Ypt31p/32p can function as upstream regulators for Vps21p.

**Vps9-CUE domain cooperates with Arf1p in Vps21p activation.** The localization of Vps21p at endosomes was significantly decreased but not completely inhibited in the *arf1Δ* mutant (Fig. 3d), suggesting that other mechanisms redundantly regulate Vps9p localization. One possible mechanism is the ubiquitin-dependent localization of Vps9p through its CUE domain that binds to mono-ubiquitinated endocytic cargo[51–53]. We found that deletion of Vps9p's CUE domain has little effect on Vps21p's localization, but results in a remarkable decrease in the fluorescent intensity of the GFP-Vps21p dots when combined with the

*arf1Δ* mutation (Fig. 6a, b). By directly comparing the *arf1Δ* mutant with the *vps9ΔCUE arf1Δ* double mutant, we found that the fluorescent intensity of GFP-Vps21p dots in the double mutant decreases compared to that seen in the *arf1Δ* mutant (Fig. 6c, d). This suggests that Vps21p activation mediated by the Vps9-CUE domain is independent of that mediated by Arf1p. The number of Vps21p-positive dots was decreased in the *vps9ΔCUE arf1Δ* mutant compared to that in *arf1Δ* cells (Fig. 6e). Pull-down analysis with GST-fused EEA1NT demonstrated that the amount of GTP-bound Vps21p does not change in the *vps9ΔCUE* mutant, but is significantly decreased in the *vps9ΔCUE arf1Δ* double mutant, although it was still higher than that seen in *vps9Δ* cells (Fig. 6f, g). A previous study reported that Arf2p, which is 96% identical to Arf1p but expressed ~10-fold lower than Arf1p, has a redundant function with Arf1p[54], and therefore, Arf2p might partially substitute for Arf1p in the *vps9ΔCUE arf1Δ* mutant.

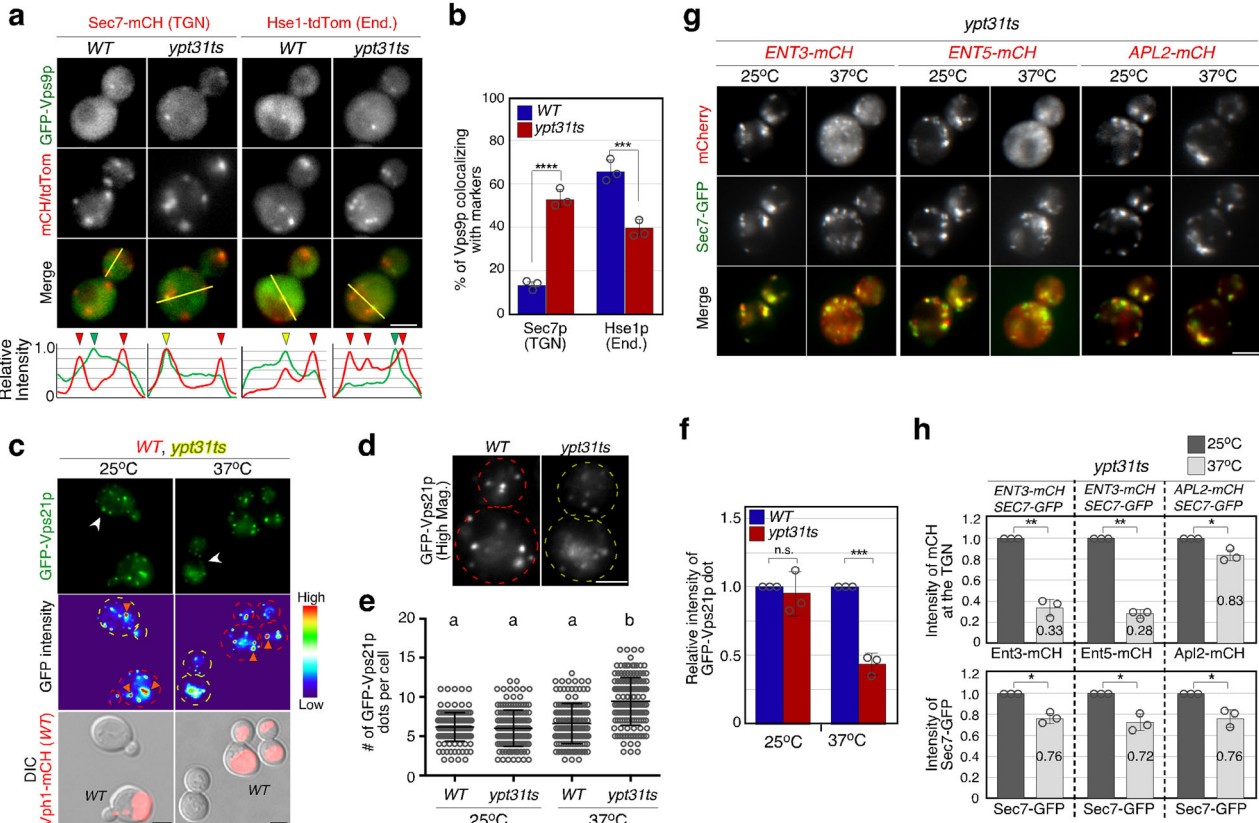

**Fig. 5** Ypt31p/32p-dependent localization of Vps9p at endosomes. **a** Colocalization of GFP-Vps9p and Sec7p-mCherry (Sec7-mCH) or Hse1p-tdTomato (Hse1-tdTom) in wild-type or *ypt31*-temperature-sensitive (*ypt31ts*) mutant cells. Ypt31p function was diminished by incubating cells at 37 °C for 2 h. Representative intensity profiles of GFP-Vps9p and Sec7-mCherry or Hse1p-tdTomato along the yellow line in the merged images are indicated in the lower graphs. Yellow or red/green arrowheads indicate the presence or absence of colocalization, respectively. **b** The percentages of colocalization were calculated as the ratio of mCH/tdTom-tagged marker (*n* = 100) colocalizing with GFP-Vps9p-positive puncta in each experiment. **c** Localization of GFP-Vps21p in the cells. Fluorescence images (GFP-Vps21p), heat maps showing GFP levels (GFP intensity) are shown. *WT* cells are labeled by the expression of Vph1-mCherry (red) which is shown in the images overlaid with DIC images. **d** High magnification images of the cells indicated with white arrows in **c** are shown. **e, f** Quantification of the number **e** or fluorescence intensity **f** of GFP-Vps21p dots displayed in **c**. **g** Localization of mCherry-tagged Ent3p (*ENT3-mCH*), Ent5p (*ENT5-mCH*), or Apl2p (*APL2-mCH*) in *ypt31ts* cells. Sec7-GFP was expressed as a control to evaluate the effect of Ypt31p dysfunction on Golgi/TGN function. **h** Quantification of the fluorescence intensity of mCherry-fused Ent3p, Ent5p, and Apl2p at the TGN (as labeled by Sec7-GFP) in *ypt31ts* cells. Intensity of Sec7-GFP was used as a control. Data show mean ± SEM with 100 puncta **b**, **f**, **h** or mean ± SD with 150 cells **e** from three independent experiments. ***$p < 0.001$, ****$p < 0.0001$, n.s., not significant, two-way ANOVA with Tukey's post-hoc test **b**. Different letters indicate significant difference at $p < 0.05$, one-way ANOVA with Tukey's post-hoc test **e** and **f**. **$p < 0.01$, n.s., not significant, unpaired *t*-test with Welch's correction **h**. Scale bar in all panels, 2.5 μm

## Discussion

EEs have been believed to be formed and maintained by the fusion of endocytic vesicles derived from the PM, and then to mature into LEs, which receive TGN-derived vesicles[3]. However, here we demonstrated that endocytic vesicle internalization is not essential for Rab5-mediated endosome formation and transport from the endosome to the vacuole. A recent study has reported that *S. cerevisiae* lacks distinct EEs and that instead the TGN is the first destination for endocytic traffic and functions as an EE-like sorting compartment[11]. This study also demonstrated that the yeast late/prevacuolar endosome is a non-maturing stable compartment[11]. In contrast to these observations, several previous studies reported that yeast display two distinct endosomes, one containing yeast Rab5 Vps21p and the other containing yeast Rab7 Ypt7p[43], and that Ypt7p replaces Vps21p during the transition from EE to LEs[55]. Additionally, we recently reported that the endocytic pathway intersects the VPS pathway from the TGN at an early stage of endocytosis, without directly contacting the TGN[15]. Thus, it has been ambiguous whether *S. cerevisiae*

contains EEs, but our findings in this study might clarify these seemingly contradictory observations.

Our data here favors the conclusion that the formation of endosomes depends on a process begun at the TGN. We demonstrated that the Rab5 GEF Vps9p is first recruited to the TGN and then transported to the endosomal compartments where it activates Vps21p (Fig. 7 and Supplementary Fig. 9). We also showed that inhibiting endocytic internalization has no effect but inhibiting post-Golgi traffic significantly reduces Rab5-mediated endosomal transport. It is, therefore, likely that post-Golgi traffic is of primary importance for endosomal formation (Fig. 7). The endocytic pathway might be only required for the transport of cargo from the PM to the TGN or for vesicles budding off from the TGN membrane. Importantly, it has been reported that Vps21p is localized at the EE-to-LEs but not at the TGN or PM[15,43], and that cells lacking Vps21p accumulate small compartments containing cargo from the endocytic and the VPS pathways in the cytosol[8,31]. Taken together, our findings suggest that recruitment of Vps9p to the TGN allows Vps21p to be

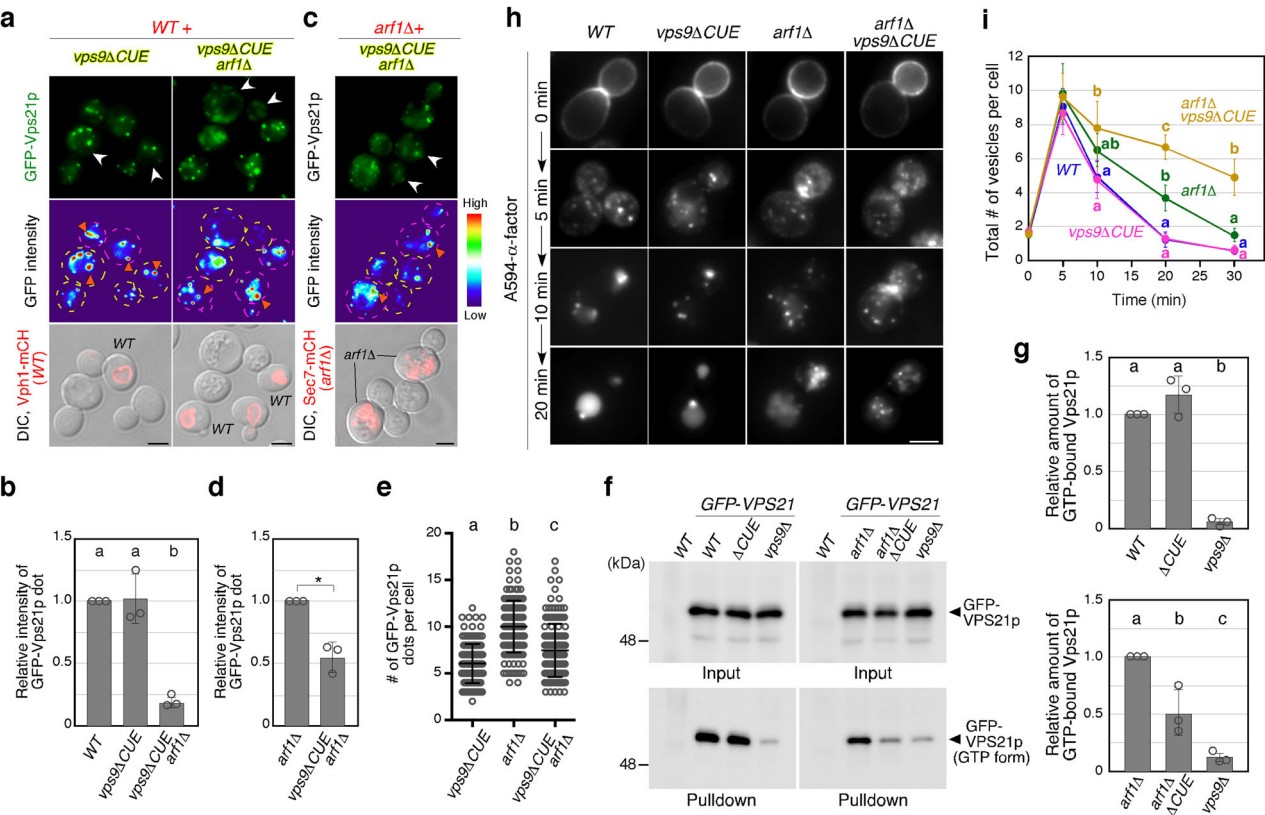

**Fig. 6** Role of Vps9's CUE domain in Vps21p-mediated endosomal formation and trafficking. **a**, **c** Localization of GFP-Vps21p in *vps9ΔCUE* and wild-type (*WT*) **a** or *arf1Δ* cells **c**. Fluorescence images (GFP-Vps21p), heat maps showing GFP levels (GFP intensity) are shown. *WT* or *arf1Δ* cells are labeled by the expression of Vph1-mCherry **a** or Sec7-mCherry **c**, respectively, which is shown in the images overlaid with DIC images. **b** and **d** Quantification of the number of GFP-Vps21p dots displayed in **a** or **c**. **e** Quantification of the fluorescence intensity of GFP-Vps21p dots displayed in **a** and **c**. **f** Immunoblots showing active levels of Vps21p. Endogenous Vps21p was tagged with GFP in the indicated cells, and 3 µg of total cell lysate (2% input) were loaded and immunoblotted with an anti-GFP antibody (GFP-Vps21p panels). Active Vps21p from 150 µg of total cell lysate was pulled down with GST-EEA1NT and probed with an anti-GFP antibody (GFP-Vps21p (GTP form) panels). **g** Graph showing mean ± SEM of the relative amount of active GFP-Vsp21p bound to GST-EEA1NT from three independent experiments. **h** Spatio-temporal localization of Alexa-α-factor in the indicated genotypes. The images were acquired at 0, 5, 10, and 20 min after washing out unbound Alexa-α-factor and incubating the cells at 25 °C. **i** Quantification of the number of Alexa-α-factor-positive vesicles displayed in **h**. Data show mean ± SD with 150 cells **e**, or mean ± SEM with 100 puncta **b**, **d** from three independent experiments. *$p < 0.05$, unpaired *t*-test with Welch's correction **d**. Different letters indicate significant difference at $p < 0.05$, one-way ANOVA with Tukey's post-hoc test **b**, **e**, and **g**, two-way ANOVA with Bonferroni's post-hoc test **i**. Scale bar in all panels, 2.5 µm. The uncropped blot for **f** can be found in Supplementary Fig. 11

recruited to the Vps9p containing TGN-derived transport vesicles, which quickly fuse with each other or with the stable pre-vacuolar endosome[15].

We also elucidated the components that regulate the delivery of the Rab5 GEF Vps9p from the TGN to the endosomes, namely two different types of TGN-resident small GTPases, Arf1p and Ypt31/32p. In this process, Arf1p and Ypt31p/32p have distinct roles: the former appears to recruit Vps9p to the TGN and the latter regulates the transport of vesicles carrying Vps9p to the endosomes via Ent3/5p (Fig. 7 and Supplementary Fig. 9). This mechanism for recruiting a Rab5 effector to the TGN to allow later Rab5 activation might be also conserved in mammalian cells. Rabaptin-5, which is a functional modulator of mammalian Rab5-GEF Rabex-5[56], associates with the TGN-residing clathrin adaptors, GGAs and the AP-1 complex[57,58]. Since Rabaptin-5 makes a tight physical complex with Rabex-5, and this complex is essential for endosomal fusion mediated by Rab5[56,57], the Rabex-5–Rabaptin-5 complex might be recruited to the TGN and then transported to the endosomes to activate Rab5.

However, a distinct clathrin complex seems to be required for Rab5 delivery in *S. cerevisiae*. We showed that among the clathrin adaptors, Ent3p/5p were specifically required for the transport of Vps9p from the TGN to the endosome. Since Ent3p and Ent5p

exhibit cargo-specific functions in trafficking proteins from the TGN to the endosome[59], Vps9p could be captured by these adaptors when vesicles bud off after being recruited to the TGN by Arf1p (Fig. 7 and Supplementary Fig. 9). Although several lines of evidence indicate that the AP-1 complex and GGAs are crucial for the function of Ent3p and Ent5p[45,60] and the localization of Ent3p at the TGN has been shown to depend on the interaction with Gga2p[60], we demonstrated that deletion of both AP-1 and GGAs has little effect on Vps21p localization on the endosomes. Thus, Ent3p and Ent5p seem to regulate Vps9p transport independently of the AP-1 complex and GGAs.

Instead Ypt31/32p recruit Ent3/5p to the TGN by a mechanism that is still unclear, but which could involve the interaction of Ent5p with phosphatydilinositol-4-phosphate (PtdIns(4)P). Previous studies reported that the levels of PtdIns(4)P at the Golgi are controlled by the TGN-localized Pik1p[59,60]. It was reported that Ent5p directly binds to PtdIns(4)P through the ANTH domain, and that depletion of PtdIns(4)P changes Ent5p localization to the cytosol, whereas overproduction of Pik1p increases the localization of Ent3p and Ent5p at the TGN[44]. Thus, PtdIns(4)P levels seem to regulate the assembly of Ent3/5p-containing clathrin-coated vesicles. Interestingly, human Rab11 (the yeast Ypt31/32p homolog) was reported to interact with PI4 kinase

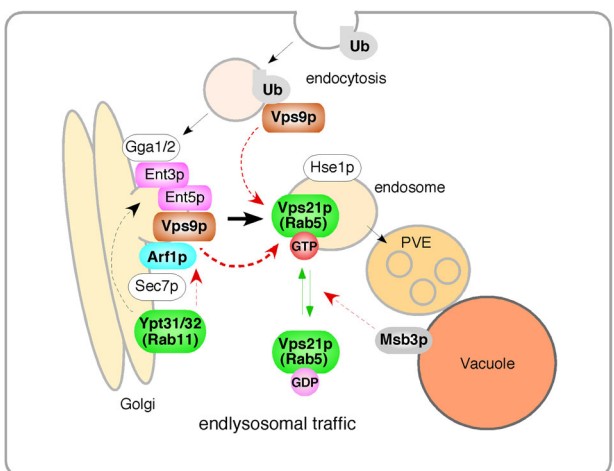

**Fig. 7** Model for the role of post-Golgi vesicle transport in Vps21p-mediated endosomal formation and trafficking. Vps9p is first recruited to the TGN and then transported to the endosomal compartments where it activates Vps21p. During this process, Arf1p directly recruits Vps9p to the TGN, and Ypt31p/32p regulates transport of Vps9p to the endosome via transport vesicles on which Ent3/5p resides. CUE domain-dependent Vps9p recruitment to the TGN or endocytic vesicles additionally regulates Vps21p activity

PI4KIIIβ (yeast Pik1p), raising the possibility that Ypt31p/32p regulate Ent3p/5p localization through an interaction with Pik1p in yeast. Studies showing that Pik1p's activity is required for the transport of internalized FM4-64 to the vacuole[61,62] and the trafficking of an ABC transporter from the endosome to the vacuole[63] also support the hypothesis that modulation of PtdIns (4)P levels, by Ypt31p/32p could be important for regulating endosomal transport through Ent3p/5p localization.

Previous studies reported the importance of ubiquitin binding by the CUE domain in the localization of Rab5 GEFs to endosomal compartments[30,64]. In mammalian cell, ubiquitin binding is necessary for the recruitment of Rabex-5, the mammalian ortholog of yeast Vps9p, to endosomes, but not sufficient because an additional interaction with the Rabaptin-5 through the C-terminal region is required[65]. A recent study also demonstrated that ubiquitin binding by the CUE domain promotes Vps9p localization at the endosome, but is not required for the biogenesis of the late endosome/multivesicular bodies[30]. Consistent with these observations, we showed that the CUE domain of Vps9p partially contributes to Vps21p activation but its role is limited. Taken together with the result that endocytosis is not essential for endosomal trafficking, CUE domain-dependent Vps9p recruitment is likely to be an additional mechanism for boosting Vps21p activation, which is constitutively regulated by the post-Golgi-dependent endosomal localization of Vps9p.

## Methods

**Yeast strains and plasmids**. The yeast strains used in this study are listed in Supplementary Table 1. All strains were grown in standard rich medium (YPD) or synthetic medium (SM) supplemented with 2% glucose and appropriate amino acids. For all experiments involving BFA treatment, the strains lacking *ERG6* gene was used to increase the permeability (Supplementary Table 1). The N-terminal GFP tag was integrated at the endogenous locus of the target gene as follows: The GFP(S65T) fragment whose stop codon was replaced with a BglII site was subcloned into BamHI-digested and NotI-digested **pBlueScript II SK** (**pBS**-GFP), and the NotI-SacII fragment, which contains the *HIS3MX6* module, was amplified by PCR using **pFA6a**-GFP (S65T)-*HIS3MX6* as a template and inserted into NotI-digested and SacII-digested **pBS**-GFP (**pBS**-GFP-HIS3). To create an integration plasmid, the promoter region of the target gene and the N-terminal fragment of the target ORF were generated by PCR and cloned into the BamHI or BglII site of **pBS**-GFP-HIS3. To integrate GFP at the N terminus of the target gene, the integration plasmid was linearized by a restriction enzyme and transformed into yeast. The N-

terminal tdTomato tag was integrated by the same procedure with the following alterations: the tdTomato fragment was amplified using **pEF1α**-tdTomato (Clontech) as a template and the *URA3MX6* or *NatMX4* module was subcloned instead of the *HIS3MX6* module. The extra region generated by the insertion of the integration plasmid was removed by PCR-based homologous recombination as shown in our previous report. C-terminal GFP or mCherry tagging of proteins was performed by PCR-based homologous recombination using **pFA6a**-GFP(S65T) or **pFA6a**-mCherry, respectively, as a template. For the analyses using the BioID method[66,67], **pFA6a**-BirA(R118G)-S tag was generated as follows: The coding sequence of BirA was obtained from *E. coli* K-12-derived XL1-Blue strain and mutated to R118G by PCR. The S-tag sequence was added to the C-terminus of BirA(R118G). The BirA(R118G)-S tag sequence was inserted into **pFA6a**-LEU2. Using **pFA6a**-BirA(R118G)-S tag-LEU2 as a template, C-terminal BirA(R118G) tagging of proteins was performed by PCR-based homologous recombination.

**Fluorescence microscopy**. Fluorescence microscopy was performed using an Olympus IX83 microscope equipped with a ×100/NA 1.40 (Olympus) objective and an Orca-R2 cooled CCD camera (Hamamatsu), using Metamorph software (Universal Imaging). Simultaneous imaging of red and green fluorescence was performed using an Olympus IX81 microscope, described above, and an image splitter (Dual-View; Optical Insights) that divided the red and green components of the images with a 565-nm dichroic mirror and passed the red component through a 630/50 nm filter and the green component through a 530/30 nm filter. Dual color time lapse imaging of red and green fluorescence was performed using an Olympus IX83 microscope equipped with a high-speed filter changer (Lambda 10-3; Shutter Instruments) that can change filter sets within 40 ms. Images for analysis of colocalization were acquired using simultaneous imaging (64.5 nm pixel size), described above. Intensity profiles of GFP-fused protein and mCherry/tdTomato-fused protein were generated across the center of fluorescence signals used for the assessment (representative intensity profiles are shown in Figs. 4g, 5a, Supplementary Figs. 1a and 3b). Colocalization was defined as occurring when the distance between the two peaks of GFP and mCherry/tdTomato intensities was <129 nm (2 pixels).

**Fluorescence labeling of α-factor and endocytosis assays**. Fluorescence labeling of α-factor was performed as described previously[34]. For endocytosis assays, cells were grown to an OD600 of ~0.5 in 0.5 ml YPD, briefly centrifuged, and resuspended in 20 μl SM with 5 μM Alexa Fluor 594-α-factor. After incubation on ice for 2 h, the cells were washed with ice-cold SM. Internalization was initiated by the addition of SM containing 4% glucose and amino acids at 25 °C.

**Pull-down assay for active Vps21p**. Recombinant GST-EEA1NT protein was expressed in *E. coli* Rosetta(DE3)pLysS (Novagen) using the **pGEX4T1** expression vector, purified using Glutathione-sepharose 4B (GE Healthcare). Cells expressing GFP-tagged Vps21p were grown in 200 ml YPD to OD600 of 1.0. The cells were harvested by centrifugation, washed with water, and resuspended in lysis buffer (20 mM Tris–HCl, pH 7.5, 100 mM NaCl, 1 mM EDTA, 10 mM DTT, protease inhibitor cocktail). Glass beads were added to an equal volume and cells were disrupted by Disruptor-Genie (Scientific industry) in the cold room. 600 μg of cleared lysates were incubated with 45–60 μg of GST-EEA1NT bound to Gluthathione-sepharose 4B for 1 h in the cold room. The sepharose was loaded into empty polypropylene column (Bio-Rad), washed three times with 5 ml lysis buffer, and bound proteins were eluted with 10 mM Glutathione. Eluted GST-Vps21p were analyzed by SDS–PAGE, followed by a western blot.

**Interaction assay using BioID**. Interactions of Vps9p with TGN-resident proteins were analyzed using proximity-dependent biotin identification (BioID) method[66,67]. Cells were grown in 200 ml SM containing 2% glucose and amino acids at 25 °C to OD$_{600}$ of 0.5–1.0. After the addition of 10 μM D-biotin, the cells were further incubated for 3 h. The cells were harvested by centrifugation, washed with water, and resuspended in lysis buffer (50 mM Tris–HCl, pH 8.0, 150 mM NaCl, 4 M Urea, protease inhibitor cocktail). Glass beads were added to an equal volume and cells were disrupted by Disruptor-Genie (Scientific industry) in the cold room. After incubation with 0.5% Triton X-100 for 10 min, 2.5 mg of the cleared lysates was incubated with Streptavidin agarose Ultra Performance (Tri-Link) for 1 h in the cold room. The agarose was washed three times with 1 ml lysis buffer containing 0.5% Triton X-100, and bound proteins were eluted with SDS–PAGE sample buffer. Eluted proteins were analyzed by SDS–PAGE, followed by a western blot.

**Western blot assay**. Immunoblot analysis was performed as described previously[68]. The rabbit polyclonal antibody to GFP (GeneTex, GTX113617) was used at a dilution of 1:10,000 and the HRP-linked donkey F(ab')$_2$ fragment to rabbit IgG (GE Healthcare, NA9340) was used as the secondary antibody at a 1:10,000 dilution. Both mouse monoclonal antibodies to GST (Cell signaling, 26H1) and GAPDH (GeneTex, GTX627408) were used at a dilution of 1:10,000 and the HRP-linked sheep F(ab')$_2$ fragment to mouse IgG (GE Healthcare, NA9310) was used as the secondary antibody at a 1:10,000 dilution. Immunoreactive proteins bands were visualized using the WesternLightning Plus ECL (PerkinElmer).

**Statistics and reproducibility**. Statistical analysis was performed with GraphPad Prism 6 software for Macintosh, and the data are shown as the mean ± S.D. or the mean ± S.E.M. as described in the figure legends. Statistical significance was determined using chi-square test for trend, unpaired *t*-test, or one-way or two-way ANOVA with post-hoc Turkey's or Bonferroni's test as described in the figure legends. For each type of analyses, at least three independent experiments were performed to confirm reproducibility.

**Reporting summary**. Further information on research design is available in the Nature Research Reporting Summary linked to this article.

## Data availability

The authors declare that all data supporting the findings of this study are available within the article and its supplementary information files. All source data underlying the graphs presented in the main or supplementary figures are made available as "Supplementary Data 1" or "Supplementary Data 2", respectively.

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

## Acknowledgements

This work was supported by JSPS KAKENHI GRANT #26440067, the Takeda Science Foundation, the Novartis Foundation (Japan) to J.Y.T., as well as JSPS KAKENHI GRANT #19K06571, the Life Science Foundation of Japan, the Uehara Memorial Foundation and the Takeda Science Foundation to J.T.

## Author contributions

Conceptualization: M.N., J.Y.T., J.T.; Methodology: M.N., J.Y.T., J.T.; Software: M.N.; Validation: J.Y.T., J.T.; Formal analysis: M.N., J.Y.T., J.T.; Investigation: M.N., J.Y.T., J.T.; Resources: M.N., J.T.; Data curation: M.N., D.E.S., J.Y.T., J.T.; Writing—original draft: J.Y.T., J.T.; Writing—review & editing: D.E.S., J.Y.T., J.T.; Visualization: D.E.S., J.Y.T., J.T.; Supervision: J.Y.T., J.T.; Funding acquisition: J.Y.T., J.T.

## Competing interests

The authors declare no competing interests.
