## [Peer Review File · Communications Biology]

Reviewers' comments:

Reviewer #1 (Remarks to the Author):

This paper uses yeast to explore the formation of endosomes. The traditional view is that endocytic vesicles fuse in a manner dependent on Rab5 (Vps21p in yeast) to generate early endosomes, which mature into late endosomes that ultimately fuse with the lysosome (or yeast vacuole). Here, evidence is presented that Vps21p-dependent endolysosomal traffic depends on post-Golgi transport but not endocytosis, and that the Vps9p protein required for activation of Vps21p is initially recruited to the TGN for transport to late endosomes with the aid of the Ent3p/5p adaptors, which in turn are recruited by the Ypt31p/32p Rab proteins. The results implicate the TGN in endolysosomal trafficking.

This work is interpreted with reference to a recent paper stating that yeast cells lack mammalian-type early endosomes and instead use TGN compartments as early/recycling endosomes. According to that concept, yeast late (prevacuolar) endosomes are long-lived structures that receive material from the TGN but do not mature from earlier endosomes. The conclusion presented here is sort of a hybrid between the two views. The TGN is acknowledged as the initial destination of endocytosis, but Vps9p is transported from the TGN to maturing endosomes to activate Vps21p.

In my opinion, the major contribution of this work is the finding that Ent3p/5p is important for Vps21p activation, and that Vps9p levels increase at the TGN while decreasing at late endosomes in an ent3-delta ent5-delta double mutant or in a ypt31-ts ypt32-delta mutant. These results will be of broad interest because they implicate a novel adaptor-mediated TGN-to-endosome transport pathway in the activation of Vps21p, which is a key regulator of endosome function.

My one significant concern is that some of the interpretations are forced. The manuscript repeatedly talks about endosomal fusion and maturation, but such phenomena are never directly observed. Instead, they are assumed to occur. That assumption is problematic because it is open to question whether endosomes in yeast actually fuse with one another in a maturation pathway. The language throughout the manuscript needs to be more rigorous in this regard. In the same vein, Figure 7 shows a model depicting a maturing endosome, but the available evidence actually argues against this idea.

For example, a section labeled "Vesicle transport from the TGN is required for Vps21p-mediated endosomal fusion." concludes with the statement: "These observations suggest that vesicle transport from the TGN is important for Vps21p-mediated endosome fusion and maturation." In fact, the results merely indicate that Golgi function is important for endocytic transport to the vacuole, a finding consistent with the view that the TGN serves as an early endosomal intermediate on the pathway to long-lived late endosomes.

Minor points:

1. In Brefeldin A-treated cells, internalized alpha-factor accumulates in punctate compartments. The text states that those structures contain Vps21p, but no data are presented. Are those structures TGN compartments, or late endosomes, or perhaps hybrid compartments?
2. The term "vesicles" usually means small spherical structures such as clathrin-coated vesicles. In this manuscript, TGN and endosomal compartments are referred to as vesicles, and this terminology may be confusing.
3. In Fig. 3, I don't understand the conclusion that deletion of Ent3p/5p causes much of the GFP-Vps21p to redistribute to the cytosol, yet the number of punctate compartments labeled with GFP-

Vps21p increases. These effects seem to be contradictory.

4. The text states that in the *ent3-delta ent5-delta* double mutant, the number of GFP-Vps21p vesicles that colocalize with Hse1-tdTomato was increased as shown in Fig. S3a,b. This statement is bizarre because according to Fig. S3b, the colocalization in WT cells is about 77 +/- 4% whereas the colocalization in the double mutant is about 79 +/- 4%. To put it mildly, that difference is not statistically significant.

Reviewer #2 (Remarks to the Author):

Nagano et al. "Rab5-mediated endosome formation is regulated at the trans-Golgi network."

In this study the authors has investigated how key proteins, Vps21p and Vps9p that mediate endosome formation are recruited onto membranes that are not derived from endocytic vesicles but are instead part of the trans-Golgi network (TGN). There has been a long held dogma that endosomes require input from endocytic vesicles along with vesicles from the TGN. This view underpins the thinking for how we understand the organisation of the endocytic system in eukaryotic cells. As such, the study from Nagano et al., challenges this view. Studies in plants have in fact blurred the lines between the early endosome and the TGN so there is some precedent to concepts being advanced by Nagano et al.

The study by Nagano et al. relies substantially on the use of yeast mutants that can block endocytosis (e.g. *end3*) and therefore if endocytosis is blocked, it logically follows that the endocytic pathway cannot provide input into the formation of endosomes positive for Vps21p or Vps9p. Nagano et al. have also made use of pharmacological inhibitors of membrane trafficking, namely Brefeldin A and Monensin to respectively inhibit the *arf1* GTPase and transport through the Golgi. The conclusions they have drawn from their data is that Vps21p (yeast Rab5) is loaded onto the TGN and that this requires *arf1* activity. Vesicles derived from the TGN that contain Vps21p can then form endosomes in the absence of any membrane from the endocytic pathway.

Overall the study contains some high quality data and there is much to commend. I do however have some reservations regarding the 'block' in endocytosis they have achieved and some concerns regarding the use of Brefeldin A and Monensin.

1. They report that endocytosis is blocked because uptake of fluorescent mating factor is blocked. But this is only one ligand and I think that to be sure that all endocytosis is blocked in their mutants, they should employ an additional endocytic ligand and also make double mutants to ensure that any parallel or redundant pathways are blocked. I'm sure the authors will be aware that screens in yeast for endocytic mutants often generate quite different results according to which ligand is used in the screen. Given that the central premise of their study is that blocking endocytosis abolishes the contribution that endocytic vesicles make towards endosome formation, it is vital that the authors demonstrate a complete and total block in endocytosis.

2. The use of Brefeldin A (BfA) and Monensin is a concern to me. It used to be the 'rule' that yeast are insensitive to BfA unless an *erg6* mutant is used. Have the authors used an *erg6* mutant for their experiments involving BfA? Why are they treating with BfA for such extended times? Generally BfA exerts its effect on *arf* proteins very quickly (usually within seconds) but in their study Nagano et al. have used BfA treatments for several minutes.

Monensin has a very undefined role in blocking membrane traffic and therefore, in my view, is not a very good inhibitor to use. I believe that Monensin is a weak base and I would therefore have concerns about its use in conjunction with FM4-64 as I think that FM4-64 requires acidity for its

fluorescence. If possible, I would urge the authors to find alternatives to the use of BfA and Monensin in their experiments as I do not think they provide clarity.

Responses to Reviewer's Specific Comments (Reviewers' comments are in italics)

Reviewer #1:

In my opinion, the major contribution of this work is the finding that Ent3p/5p is important for Vps21p activation, and that Vps9p levels increase at the TGN while decreasing at late endosomes in an ent3-delta ent5-delta double mutant or in a ypt31-ts ypt32-delta mutant. These results will be of broad interest because they implicate a novel adaptor-mediated TGN-to-endosome transport pathway in the activation of Vps21p, which is a key regulator of endosome function.

We are very grateful for this reviewer's favorable evaluation of our studies.

Comment #1

My one significant concern is that some of the interpretations are forced. The manuscript repeatedly talks about endosomal fusion and maturation, but such phenomena are never directly observed. Instead, they are assumed to occur. That assumption is problematic because it is open to question whether endosomes in yeast actually fuse with one another in a maturation pathway. The language throughout the manuscript needs to be more rigorous in this regard. In the same vein, Figure 7 shows a model depicting a maturing endosome, but the available evidence actually argues against this idea.

In accordance with the reviewer's suggestion, we have changed all the references to "endosomal fusion and maturation" included in the previous manuscript to more suitable terms, such as endosomal transport. We have also deleted a maturing endosome from the model in Fig. 7 in the new manuscript.

For example, a section labeled "Vesicle transport from the TGN is required for Vps21p-mediated endosomal fusion." concludes with the statement: "These observations suggest that vesicle transport from the TGN is important for Vps21p-mediated endosome fusion and maturation." In fact, the results merely indicate that Golgi function is important for endocytic transport to the vacuole, a finding consistent with the view that the TGN serves as an early endosomal intermediate on the pathway to long-lived late endosomes.

We have changed the sentence "~ Vps21p-mediated endosome fusion and maturation." to "~ Vps21p-mediated endosomal transport to the vacuole." in the new manuscript.

Minor points:

#1. *In Brefeldin A-treated cells, internalized alpha-factor accumulates in punctate compartments. The text states that those structures contain Vps21p, but no data are presented. Are those structures TGN compartments, or late endosomes, or perhaps hybrid compartments?*

In the previous manuscript, we showed that internalized alpha-factor accumulates in punctate compartments containing Vps21p in Monensin-treated cells, but did not show it

in BFA-treated cells. In the new manuscript, we have added data showing that BFA treatment induces accumulation of internalized alpha-factor in puncta containing Vps21p (Fig. S2a-c).

We also examined the Vps21p localization in BFA-treated cells and found that GFP-Vps21p is well colocalized with an endosomal marker, Hse1-mCherry, but not with the TGN marker, Sec7-mCherry (Fig. S2d,e). Therefore, we concluded that in BFA or Monensin-treated cell, internalized alpha-factor accumulates in endosomal compartments localizing GFP-Vps21p.

#2. The term “vesicles” usually means small spherical structures such as clathrin-coated vesicles. In this manuscript, TGN and endosomal compartments are referred to as vesicles, and this terminology may be confusing.

According to reviewer’s suggestion, we have changed the term to more suitable words, such as dots, puncta and endosomes. For instance, in the sentences regarding Fig. 1, 2a,b and 3, we have changed the term “vesicles” to “endosomes” because we have showed that GFP-Vps21p-labeled compartments are endosomes (Fig. S1c,d, S2d,e and S5a,b). In Fig. 2d and S3e,f, we have changed the term to “puncta” because we have not determined which compartments are labeled by FM4-64.

#3. In Fig. 3, I don’t understand the conclusion that deletion of Ent3p/5p causes much of the GFP-Vps21p to redistribute to the cytosol, yet the number of punctate compartments labeled with GFP-Vps21p increases. These effects seem to be contradictory.

We agree with the reviewer’s comment; our explanation was insufficient to make the reader understand our interpretation of the results correctly. We speculate that in the *ent3Δ ent5Δ* mutant activation of Vps21p is partially impaired and Vps21p cannot be recruited to the endosome from the cytosol. As a result, decreased endosomal localization of activated Vps21p reduces the competence of endosomal transport, causing an increase of punctate compartments labeled with GFP-Vps21p in this mutant. To explain about this, we have added some sentences in the results section (line 235-242 in the new manuscript).

#4. The text states that in the ent3-delta ent5-delta double mutant, the number of GFP-Vps21p vesicles that colocalize with Hse1-tdTomato was increased as shown in Fig. S3a,b. This statement is bizarre because according to Fig. S3b, the colocalization in WT cells is about 77 +/- 4% whereas the colocalization in the double mutant is about 79 +/- 4%. To put it mildly, that difference is not statistically significant.

We appreciate the reviewer’s point. We did not explain the results shown in Fig. 3a-d correctly in the previous manuscript. We wanted to explain that the number of GFP-Vps21p vesicles (endosomes) was increased, and also explain that colocalization of GFP-Vps21p with Hse1p was not changed in the mutant, compared with wild-type cell, indicating that GFP-Vps21p is still localized at the endosomes in the mutant. To correctly

explain about these results, we have added some sentences in the results section (line 235-239 in the new manuscript).

Reviewer #2 (Remarks to the Author):

Comment #1

They report that endocytosis is blocked because uptake of fluorescent mating factor is blocked. But this is only one ligand and I think that to be sure that all endocytosis is blocked in their mutants, they should employ an additional endocytic ligand and also make double mutants to ensure that any parallel or redundant pathways are blocked. I'm sure the authors will be aware that screens in yeast for endocytic mutants often generate quite different results according to which ligand is used in the screen. Given that the central premise of their study is that blocking endocytosis abolishes the contribution that endocytic vesicles make towards endosome formation, it is vital that the authors demonstrate a complete and total block in endocytosis.

We agree with the reviewer's opinion that it is vital that we demonstrate a complete and total block in endocytosis in mutants we used in Fig.1. In accordance with the reviewers' suggestion, we performed three experiments in the new manuscript. First, we additionally utilized a lipophilic styryl dye, FM4-64, that is used to follow bulk membrane, and showed that the *sac6Δ scp1Δ* mutant showed a remarkable defect in the uptake of FM4-64 from the plasma membrane (Fig. S1a,b). Second, we utilized the *sac6Δ scp1Δ rom1Δ* triple mutants to block a redundant endocytic pathway. A previous study by Dr Wendland's group reported that yeast has a clathrin-independent endocytic pathway that depends on the Rho1 GTPase (Prosser et al., JCB, 2011). Thus, we generated triple mutant lacking the Rho1-GEF gene, *ROM1*, in addition to the *SAC6* and *SCP1* genes. The *sac6Δ scp1Δ rom1Δ* triple mutant exhibited a severer defect in FM4-64 uptake from the plasma membrane, compared to the *sac6Δ scp1Δ* double mutants (Fig. S1a,b), but the localization of Vps21p was almost the same as that in wild-type cells (Fig. 1c-e). Finally, we examined the effect of Latrunculin A, which inhibits both clathrin-dependent and -independent endocytosis, on Vps21p localization, and found that the Latrunculin A treatment did not affect Vps21p localization. These results support our conclusion that endocytic vesicle internalization is not essential for Rab5-mediated endosome transport.

Comment #2

*The use of Brefeldin A (BfA) and Monensin is a concern to me. It used to be the 'rule' that yeast are insensitive to BfA unless an *erg6* mutant is used. Have the authors used an *erg6* mutant for their experiments involving BfA?*

As the reviewer inferred, we have used strains lacking *ERG6* gene for all the experiments involving BFA treatment. We have now added the sentence “For all experiments involving BFA treatment, strains lacking the *ERG6* gene was used to increase permeability (Table S1).” in “Yeast strains and plasmids” section of “Materials and Methods” in the new manuscript.

Why are they treating with BfA for such extended times? Generally BfA exerts its effect on arf proteins very quickly (usually within seconds) but in their study Nagano et al. have used BfA treatments for several minutes.

We have added data showing the short-term effects of BFA treatment on the localization of Arf1p and GFP-Vps21p in Fig. S4c. As the reviewer suggested, we found that BFA treatment changed Arf1p localization very quickly (within 5 min), and also induces the accumulation of Vps21p with the same timing (Fig. S4c). We had initially speculated that when cells were treated with BFA, although Vps21p activation is inhibited, Vps21p, which had been already activated and localized to endosomes, is transported to the prevacuolar endosome/vacuole, and then disperses in the cytosol by being inactivated by its GAP Msb3p. Thus, it might take more than 30 minutes to substantially change the Vps21p localization to the cytosol. Actually, we examined the fluorescent intensity of GFP-Vps21p in the cytosol at 5-10 min after the BFA treatment, but we were not able to detect the significant change in the intensity (Fig. S4c). In the new manuscript, we have added the data showing the short-term effects of BFA treatment in Fig. S4c.

Monensin has a very undefined role in blocking membrane traffic and therefore, in my view, is not a very good inhibitor to use. I believe that Monensin is a weak base and I would therefore have concerns about its use in conjunction with FM4-64 as I think that FM4-64 requires acidity for its fluorescence. If possible, I would urge the authors to find alternatives to the use of BfA and Monensin in their experiments as I do not think they provide clarity.

To resolve the reviewer’s concern about use of Monensin with FM4-64, we have additionally examined the effect of BFA treatment on FM4-64 uptake and found that BFA treatment causes a similar delay in FM4-64 uptake. In the new manuscript, we have replaced Fig. 2d,e from the data using Monensin to that using BFA, and moved the original Fig. 2d,e (data using Monensin) to Fig. S3e,f.

We agree with the reviewer’s comment that BFA and Monensin are not a very good inhibitor to use, but we were not able to find a better alternative drug that can specifically inhibit post-Golgi transport to the vacuole. However, to support the data obtained from the experiment using these drugs, we have shown data using several mutants, including *arf1Δ*, *ent3Δ ent5Δ*, and *ypt3Its* mutants that have defects in post Golgi vesicle transport, in Fig. 3-5. We believe that our conclusion was strengthened by combining the experiments using drugs with those using these mutant strains.

REVIEWERS' COMMENTS:

Reviewer #1 (Remarks to the Author):

The authors have improved the paper. The changes made in response to my suggestions are satisfactory. It's an interesting story, and I have no further comments.

Reviewer #2 (Remarks to the Author):

The manuscript has been improved with additional data to address the comments and concerns I raised previously. On balance I think I can now recommend publication.